# Research on Point Cloud Acquisition and Calibration of Deep Hole Inner Surfaces Based on Collimated Ring Laser Beams

**DOI:** 10.3390/s24175790

**Published:** 2024-09-06

**Authors:** Huifu Du, Xiaowei Zhao, Daguo Yu, Hongyan Shi, Ziyang Zhou

**Affiliations:** 1School of Mechanical Engineering, North University of China, Taiyuan 030051, China; b20220206@st.nuc.edu.cn (X.Z.); yudaguo@nuc.edu.cn (D.Y.); 2202041109@st.nuc.edu.cn (Z.Z.); 2Shanxi Deep Hole Processing Engineering Technology Research Center, Taiyuan 030051, China; 3Department of Mechanical Engineering, Shanxi Engineering Vocational College, Taiyuan 030051, China; shihongyan@sxgy.edu.cn

**Keywords:** deep hole detection, ring beam, collimated beam, 3D reconstruction, point cloud calibration

## Abstract

In this study, a ring light point cloud calibration technique based on collimated laser beams is developed, aiming to reduce errors caused by the position and attitude changes of traditional ring light measurement devices. This article details the generation mechanism of the ring beam and the principle of deep hole measurement. It introduces the collimated beam as a reference, building on traditional ring light measurement devices, to achieve the synchronous acquisition of the ring beam and collimated spot images by an industrial camera. The Steger algorithm is employed to accurately extract the coordinates of the point cloud contours of both the ring beam and the collimated spot. By analyzing the shape and position changes of the collimated spot contour, the spatial position and attitude of the measuring device are precisely determined. This technique is applied to the 3D reconstruction of the inner surface of deep holes, ensuring the accurate restoration of the spatial positional attitude of the ring beam by incorporating the spatial positional attitude parameters of the measuring device to precisely calibrate the cross-sectional point cloud coordinates. Experimental results with ring gauges and deep hole workpieces demonstrate that this technique effectively reduces the percentage of point cloud data outside the tolerance range, and improves the accuracy of the 3D reconstruction model by 6.287%, thereby verifying the accuracy and practicality of this technique.

## 1. Introduction

In industrial manufacturing, accurately measuring the internal profile and geometric parameters of deep hole workpieces is essential for ensuring product quality and performance, particularly in the automotive, aerospace, and heavy industries. Precise detection of internal structures in deep holes, such as engine cylinders, hydraulic cylinders, pipelines, and artillery barrels, is of particular importance [1,2]. In recent years, researchers have proposed various detection methods and developed reliable commercial measurement instruments [3,4]. These instruments are primarily categorized into contact and non-contact measurements. In contact measurement, coordinate measuring machines (CMM) represent a high-precision detection method; however, they are costly, time-consuming, and incapable of measuring deep holes with large depth-to-diameter ratios. Other contact measurement instruments, such as internal diameter dial gauges, calipers, and gauge blocks, are less expensive but offer relatively lower measurement accuracy and are generally limited to single-parameter measurements.

With the rapid development of optical measurement technology, non-contact optical measurement methods have become important tools in the field of industrial inspection [5,6]. Among these, ring beam detection technology has shown significant advantages in deep hole detection due to its non-contact, fast, high-precision, and high-resolution characteristics, attracting widespread attention [7]. Ring beam technology utilizes a specially designed conical mirror to convert a laser into a light plane. When this light plane intersects with the hole wall, it forms a ring beam. The image of the ring beam is captured by a CCD or CMOS camera, and the ring contour coordinates are extracted through image processing algorithms, thereby accurately obtaining the profile information of the internal contour. However, existing technologies have not sufficiently addressed the issues of position and attitude changes during the measurement process. Yoshizawa and Wakayama’s research team studied a method of measuring the inner diameter and contour of pipes and holes using a ring beam device, and analyzed the center alignment technique of the ring beam device [8,9,10]. They proposed a method for simultaneously measuring internal and external contours using a ring beam device [11,12,13], and developed a compact measurement instrument for measuring the internal contours of tubular objects. Zhao [14] designed a multi-diameter concentric calibration gauge to calibrate the system’s optical plane, addressing the issues of optical plane deviation and one-dimensional linear motion errors in traditional calibration processes. Dong [15] proposed an inner hole measurement method based on optical vision measurement principles. The measurement system is divided into three subsystems, providing global alignment information, axial depth information, and inner surface profile information, enabling the reconstruction of a three-dimensional model of the track. Dong [16] proposed a new two-step calibration method to compensate for system errors caused by optical plane deviation and moving stage motion direction deviation. Buschinelli [17] developed an optical measurement system based on laser conical triangulation for detecting the internal geometry of corroded pipes. Li [18] developed a multi-ring structured light system to obtain internal contour information of deep holes such as inner diameters, and designed a distortion correction method using a two-step calibration technique to improve the multi-ring structured light measurement system. Zhu [19,20] studied inner surface detection technology based on circular structured light. By using binocular vision and a flexible pre-calibrated camera method, they achieved high-precision inner surface detection without the need for precise assembly. Most researchers have deeply explored the state where the ring beam cross-section (ring beam plane) is perpendicular to the deep hole axis, capturing ring beam contour images with a camera, and converting the contours into point cloud data through image processing algorithms, focusing on improving the measurement accuracy of single cross-sectional data [21]. Ideally, evenly spaced stitching of multiple cross-sectional point cloud data can achieve three-dimensional reconstruction of the inner surface of deep holes. Based on point cloud data, geometric parameters such as inner diameter, straightness, cylindricity, and roundness, along with defect detection, can be achieved [22]. However, in the actual measurement process of deep holes, the position and attitude of the ring beam device change as it moves within the deep hole cavity. This results in misalignment between the ring beam cross-section and the deep hole workpiece axis. Using evenly spaced stitching of cross-sectional point clouds can lead to errors in the three-dimensional reconstruction of the inner surface of deep holes, directly affecting the accuracy and reliability of the measurements.

In addition, there are some methods with higher accuracy. López [23] developed a holographic interferometer-based method for cylindrical surface measurement, which can achieve an optimal sensitivity of 6.25 µm. Yokota [24] also used holographic interferometry to measure the inner wall of a pipeline and applied synthetic wavelength to perform measurements in the height ranges, exceeding the single laser wavelengths. This approach requires the very stable two-wavelength source; thus, the tilt angle illumination method can be an effective alternative [25] with lower costs. These methods can achieve high accuracy when the target surface is smooth enough, but they may fail when the target surface is too rough to form clear and continuous interference fringes. Arrizubieta [26] used a confocal microscope for hole internal shape characterization during laser percussion drilling, but it is relatively expensive and usually does not have enough measuring range for deep holes with larger diameters. In view of this, this study proposes a collimated laser beam-based point cloud calibration method for deep hole internal surface cross-sections, which optimizes and improves the mechanical structure of the traditional ring beam device. By adopting a collimated laser beam as the measurement reference, a collimated spot is generated by irradiating it onto the optical screen surface of the measurement device, and the camera not only collects the image of the ring beam but also collects the image of the collimated spot synchronously during the measurement process. By analyzing the changes in the shape and position of the collimated spot, the position and attitude angle of the measuring device can be effectively solved. On the basis of equidistant splicing of the coordinates of point clouds from multiple cross-sections, the position and attitude direction of each cross-section are introduced for accurate correction to ensure that high-quality point cloud data are obtained, which significantly improves the evaluation accuracy of deep hole parts.

This study not only deepens the exploration and optimization of ring beam technology in the measurement of deep hole workpieces but also significantly enhances the accuracy and reliability of measurement data by overcoming the negative impact of changes in the position and attitude of the measurement device on measurement accuracy. The research results are expected to provide a solid theoretical and practical foundation for the further application and development of ring beam detection technology. Through this study, it is hoped to provide a faster, safer, and more accurate method for the internal measurement of deep hole workpieces, offering a scientific basis and technical support for quality control and safety inspection in related industries.

## 2. Basic Principles of Deep Hole Detection

Figure 1 illustrates the diagram of ring beam deep hole detection principles. The measurement system primarily consists of three parts: the ring beam measurement device, the power drive device, and the laser collimation device. The ring beam measurement device includes a camera, a 90° conical mirror, a glass tube, an acrylic plate, a light screen, and a laser; the power drive device is an electric cylinder; and the laser collimation device is a collimated laser placed outside the deep hole workpiece. The green sections in Figure 1 and Figure 2 represent the ring beam. The laser and the conical mirror are installed at the center of the glass tube, with the straight laser beam generated by the laser directly illuminating the top of the conical mirror. The laser beam reflects off the conical surface, forming a 360-degree omnidirectional laser beam, which can penetrate the glass tube and directly illuminate the inner wall of the deep hole. The blue section in Figure 2 represents the inner wall of the deep hole. When the 360-degree omnidirectional laser beam intersects with the inner wall, it generates ring laser fringes, which are defined as the ring beam. When this beam is projected onto the inner wall, the light undergoes both specular and diffuse reflection on the surface. The camera is set on one side of the beam and is able to capture these reflected rays through the acrylic plate, thereby obtaining an image of the distribution of the laser fringes on the inner wall. The ring beam measurement device is mounted at the end of the electric cylinder. This study employs the Steger [27,28,29] algorithm for image processing to precisely extract the centerline contour of the ring beam, thereby obtaining the point cloud coordinates of the deep hole cross-section. By controlling the electric cylinder to drive the ring beam measurement device to move within the deep hole cavity, it is equivalent to laser scanning to acquire point cloud coordinates of various cross-sections. Finally, multiple cross-sectional point cloud coordinates are spliced and registered to construct a complete three-dimensional point cloud model of the deep hole’s inner surface.

In the deep hole measurement scheme proposed in this study, a motorized telescopic cylinder drives the ring light measurement device to move in the inner cavity of the deep hole. Since the motorized cylinder belongs to the cantilever beam structure, the position and attitude of the ring light measurement device will change with the increase in the extension. Figure 3 shows the scanning trajectory of the ring beam plane in the deep bore inner cavity, and the spatial right-angle coordinate system is established with the deep bore axis as the Z-axis. Based on the detection principle, the optical axis of the laser does not need to be strictly aligned with the axis of the deep hole. It is only necessary to ensure that the changes in the ring beam during the measurement process remain within the camera’s FOV (field of view). This is a major advantage of the ring beam measurement device. At the initial position, the optical axis of the optical measurement device is set near the axis of the deep hole, ensuring they coincide as much as possible, and the coordinate point of the light source center of the ring beam plane is O (X_start_, Y_start_, Z_start_), at which time the image of the ring beam acquired by the camera is an ideal circle. With the increase in the extension length of the electric cylinder, the spatial position and attitude of the optical measuring device change, and the coordinate point of the center of the light source in the ring beam plane changes to the position of O′ (X_end_, Y_end_, Z_end_), at which time the ring beam acquired by the camera changes from a circle to an ellipse. In the process of splicing the point cloud of each cross-section, it is necessary to accurately describe the geometric properties of the ring beam of each cross-section in space to ensure the accuracy of 3D reconstruction.

This study introduces a novel method for determining the spatial position and attitude parameters of ring beams using a collimated laser beam. As shown in Figure 2, this method integrates a laser collimation device into the traditional ring beam measurement apparatus. A translucent aerospace nano-film is adhered to one side of the acrylic plate to form a light screen. Its function is to receive the collimated laser beam and generate a collimated light spot on its surface, enabling the camera to capture clear images of the collimated light spot. The collimated laser is positioned outside the deep hole workpiece. It produces a uniform parallel beam with a constant diameter of 10 mm. This beam can pass through the acrylic plate and project onto the light screen of the measurement device, forming a collimated light spot. The electric cylinder, V-block, and collimated laser are all mounted on the measurement platform. The ring beam measurement device is installed at the end of the electric cylinder, and the collimated laser beam, serving as a measurement reference, continuously illuminates the light screen of the measurement device. During the measurement process, the electric cylinder drives the ring beam measurement device to move within the deep hole cavity. The camera not only captures images of the ring beam but also simultaneously captures images of the collimated light spot. If the position and shape of the collimated spot change during the measurement, this indicates a change in the spatial position and attitude of the ring beam measurement device. By analyzing the changes in the center point and elliptical contour shape of the collimated light spot, the spatial position and orientation parameters of the optical measurement device can be determined. When performing three-dimensional reconstruction of the deep hole’s inner surface, the position and orientation information of the optical measurement device is incorporated. This allows for precise calibration of the point cloud coordinates of each cross-sectional ring beam, ensuring accurate restoration of the ring beam’s spatial position and orientation, thereby improving the accuracy of point cloud data acquisition. Based on the calibrated point cloud data of the deep hole’s inner surface, various algorithms can be applied to analyze geometric parameters and surface defect information. These algorithms include the least squares method [30], genetic algorithm [31], minimum circumscribed circle and maximum inscribed circle of roundness deviation [32], and Canny edge detection [33]. These methods can be employed to evaluate collective parameters such as the deep hole’s inner diameter, roundness, and straightness, and to detect and analyze information about defects on the inner surface.

The advantage of this approach lies in its ability to simultaneously process the contours of the ring beam and collimated spot within the same image. Changes in the shape and position of the collimated spot contour can reflect the spatial position and attitude of the measurement device in real time, thereby providing calibration data for the ring beam point cloud coordinates. This measurement principle is also applicable to the inspection of deep holes with extremely high length-to-diameter ratios. By integrating the optical measurement device into a pipeline robot, replacing the electric cylinder with this robot for actuation, and using the collimated laser beam as a measurement reference, comprehensive, high-precision, and efficient data collection of the inner walls of deep holes can be achieved.

## 3. Mathematical Model for Calibration of Ring Beam Cross-Sectional Point Clouds

Based on the measuring principle, the trajectory of the optical measuring device during measurement is similar to the trajectory of the deflection curve equation formed by a cantilever beam with a downward force applied at the free end. The attitude position of the measuring device can be determined by measuring the direction and the radial offset distance.

### 3.1. Measurement Direction Vector Solution for a Ring Beam Measurement Device

The collimated beam is irradiated onto the screen to form a spot, which is equivalent to cutting the collimated laser beam on the light screen. In the ideal measurement situation, the optical measuring device moves axially under the drive of the electric cylinder, and the outline of the collimated spot is a standard circle. However, in the actual measurement, due to the change of the positional attitude of the optical measuring device, the outline shape of the collimated spot changes from a circle to an ellipse. As shown in Figure 4, a three-dimensional model illustrates the collimated spot change process, in which the cylinder represents the collimated laser beam, and the blue area is the collimated spot formed by the intersection of the light screen and the collimated laser beam. In the plane of the collimated spot, the spatial coordinate system O-XYZ is established with the axis of the collimated laser beam as the Z-axis. Firstly, the measuring device is rotated by α around the Y-axis to generate the new coordinate system O-X′Y′Z″, and, secondly, the measuring device is rotated by β around the X′-axis to generate the spatial coordinate system O-X″Y″Z″, the new X″-axis is connected with the plane of the collimated spot to generate the spatial coordinate system O-X″Y″Z″, and the new Z″ axis is perpendicular to the plane of the collimated spot. The spatial attitude change of the measuring device can be decomposed into the above two spatial coordinate system transformations, during which the shape of the collimated spot changes from a circle to an ellipse.

The elliptical shape change is related to the attitude angle of the optical measuring device. The elliptical change of the collimated spot is geometrically analyzed, as shown in Figure 4, and projected along the direction of the two rotational axes (in the direction of the red arrows). It is found that two right-angled triangles can be constructed in the projected view RT ΔONX_0_, RT ΔOMY_0_. Based on the plane geometry, two angles of rotation of the measuring device can be obtained in these two triangles:(1)$$\alpha =\mathrm{arc}\cos\frac{O^{\prime \prime }N}{O^{\prime \prime }X_{0}}$$
(2)$$\beta =arcsin\left(\sqrt{\frac{1-\frac{R^{2}}{{OM}^{2}}}{cos^{2}\alpha }}\right)$$
where OM and ON are the intercepts of the elliptic curve on the X and Y axes, respectively, and O″X_0_ and R are the radii of the collimated beam.

The elliptic parameters can be obtained by fitting the point cloud coordinates of the collimated spot profile, and the above parameters are known quantities, from which the two rotation angles α and β of the optical measuring device can be resolved, where α can be defined as the yaw angle and β can be defined as the pitch angle, which can determine the measurement direction vector of the ring optical measuring device.

### 3.2. Radial Offset Position Solution for Ring Optical Measurement Device

Figure 5 illustrates the camera pinhole imaging model, where the camera always acquires images perpendicular to the normal direction of the ring light section. In the pixel coordinate system, the camera acquires images of the ring beam at different positions, and the extracted contour point cloud coordinates will be different. In the figure, D represents the contour diameter of the ring optical cross-section, the initial position of the pinhole is at point A, and D′ represents the projection of the optical cross-sectional diameter on the camera imaging surface. Assuming that the camera is offset downward from the laser, the initial position A of the pinhole is offset downward to point B. At this time, the diameter of the ring optical cross-section in the camera imaging plane is D″. The change in the optical cross-sectional profile in the camera imaging plane is analyzed. Although the optical cross-sectional diameter D″ is the same length as D′, the coordinate points of the cross-sectional profile produce a relative offset, which can be quantified by the offset distance P of the coordinate point at the center of the profile.

Based on the above analysis, Figure 6a draws a schematic diagram of the radial offset of the optical measurement device, in which the ring beam and the collimated spot are located in the same plane, assuming that the initial position of the optical measurement device is at point A. The device undergoes a radial (Y-axis direction) offset during the driving process, and the offset trajectory moves from point A to point D. Figure 6b shows the trajectory of the ring beam and the collimated spot profile in the pixel coordinate system. At the initial position, the center of the ring beam is at position A′ and the center coordinate of the collimated spot is at position A″. When the optical measuring device is radially offset, the position of the circular center of the ring beam and the position of the center of the collimated spot are changed, and the coordinates of the profiles of both the ring beam and the collimated spot are offset by the same distance in the y-direction with respect to the initial position. The contour of the ring laser beam contains information about the inner wall of the deep hole, and the relative positions of the circular center coordinates of the ring beam of each cross-section cannot be used for point cloud calibration during the 3D reconstruction process of the point cloud stitching of each cross-section. However, the relative position of the circular center coordinates of the collimated light spots of each cross-section is determined, and the variation of this circular center coordinate is not subject to external interference. Therefore, the variation of the circular center coordinates of the collimated light spots can be mapped to the variation of the circular center coordinates of the ring beam. Using the circular center coordinate A″ of the collimated light spot of the first cross-section as a reference, the point cloud coordinates of each of the other cross-sections are shifted A″B″, A″C″, and A″D″ distances, which can compensate for the effect of the radial offset of the optical measuring device, thereby realizing accurate point cloud of each cross-sectional splicing.

When the optical cross-section does not coincide with the deep hole axis, the collimated spot profile image captured by the camera changes from a circle to an ellipse, and the above method is still applicable. The relative change of the ellipse center can be utilized to compensate for the radial offset error of the optical measuring device, further realizing the calibration of the point cloud of each cross-section.

### 3.3. Calibration of Point Clouds in the Ring Beam Cross-Section

This section details the methodology for fitting the elliptical contour of the collimated spot and determining the positional and attitudinal parameters of the optical measurement device.

#### 3.3.1. Determination of the Elliptical Parameters of the Collimated Spot

Based on the analysis in Section 3.1 and Section 3.2 above, the shape and positional variations of the collimated spot contour are the key parameters for calibrating the cross-sectional point cloud. By performing image processing on the collimated spot profile image, the point cloud coordinates of the profile plane can be obtained at the sub-pixel level. Since the collimated spot profile exhibits elliptical variations during the measurement process, the following elliptic function is used for regression analysis:(3)$$ax^{2}+bxy+cy^{2}+dx+ey+1=0$$

Definition of the minimum energy function:$$F=\sum {\left(a{x_{i}}^{2}+bx_{i}y_{i}+c{y_{i}}^{2}+dx_{i}+ey_{i}\right)}^{2},i\in [1,N]$$
where N denotes the number of points in the cross-sectional point cloud. Its minimum value is solved according to the method of least squares; function F is derived and converted to matrix form:$$\left[\begin{array}{ccccc} \Sigma \left(x_{i}^{4}\right) & \Sigma \left(x_{i}^{3}y_{i}\right) & \Sigma \left(x_{i}^{2}y_{i}^{2}\right) & \Sigma \left(x_{i}^{3}\right) & \Sigma \left(y_{i}x_{i}^{2}\right)\\ \Sigma \left(x_{i}^{2}y_{i}\right) & \Sigma \left(x_{i}^{2}y_{i}^{2}\right) & \Sigma \left(x_{i}y_{i}^{3}\right) & \Sigma \left(x_{i}^{2}y_{i}\right) & \Sigma \left(x_{i}y_{i}^{2}\right)\\ \Sigma \left(x_{i}^{2}y_{i}^{2}\right) & \Sigma \left(x_{i}y_{i}^{3}\right) & \Sigma \left({y_{i}}^{\;2}\right) & \Sigma \left(x_{i}y_{i}^{2}\right) & \Sigma \left(y_{i}^{3}\right)\\ \Sigma \left(x_{i}^{2}\right) & \Sigma \left(x_{i}^{2}y_{i}\right) & \Sigma \left(x_{i}y_{i}^{2}\right) & \Sigma \left(x_{i}^{2}\right) & \Sigma \left(x_{i}y_{i}\right)\\ \Sigma \left(y_{i}x_{i}^{2}\right) & \Sigma \left(x_{i}y_{i}^{2}\right) & \Sigma \left(y_{i}^{3}\right) & \Sigma \left(x_{i}y_{i}\right) & \Sigma \left(y_{i}^{2}\right) \end{array}\right]\left[\begin{array}{c} a\\ b\\ c\\ d\\ e \end{array}\right]=\left[\begin{array}{c} -\Sigma \left({x_{i}}^{2}\right)\\ -\Sigma \left(x_{i}y_{i}\right)\\ -\Sigma \left({y_{i}}^{2}\right)\\ -\Sigma \left(x_{i}\right)\\ -\Sigma \left(y_{i}\right) \end{array}\right]$$

By solving the above equations, the center coordinate point of the elliptical profile of the collimated spot can be obtained, along with the intercept of the elliptic curve equation on the X and Y axes.

#### 3.3.2. Calibration of Radial Offset Errors in Point Clouds of the Ring Beam Cross-Section

Following the analysis presented, the changes in the coordinates of the ring beam points due to the radial offset of the optical measurement device can be quantified by the shifts in the elliptical center coordinates of the collimated spot contour. As discussed earlier, by conducting elliptical fitting on the collimated spot contour point cloud coordinates across n cross-sections, the elliptical center coordinates (X_oi_, Y_oi_) for each of the n cross-sections can be determined, where i = 1, 2, …, *n*. By using the center coordinate point (X_o1_, Y_o1_) from the first cross-section as a reference, translating the elliptical center coordinates of subsequent cross-sections to this reference position results in a corresponding translation of the point cloud coordinates (x_i_, y_i_) for each cross-section, thereby generating new cross-section point clouds:(4)$$\left[\begin{array}{l} x_{i}^{\prime }\\ y_{i}^{\prime } \end{array}\right]=\left[\begin{array}{l} x_{i}+a_{i}\\ y_{i}+b_{i} \end{array}\right]$$
where x_i_ represents the x-coordinate data set of the cross-section point cloud, and y_i_ represents the y-coordinate data set of the cross-section point cloud. $a_{i}=x_{oi}-x_{o1}$, $b_{i}=y_{oi}-y_{o1}$, i=2,3,4….

Based on the measurement principle, the 3D reconstruction of the inner surface of a deep hole requires the alignment of multiple cross-sectional point clouds. Using the center of the collimated spot of the first cross-section as a reference, the ring beam contour coordinates of the remaining cross-sections can be translated according to Equation (4) to compensate for the point cloud stitching errors caused by the radial offset of the optical measurement device.

#### 3.3.3. Calibration of Rotational Errors in Point Clouds of the Ring Beam Cross-Section

The point cloud error generated by the radial offset of the optical measuring device is calibrated as described above, on the basis of which the rotational calibration of the point cloud of the ring beam cross-section is carried out. According to Equations (1) and (2), it is known that under the parameter conditions of the elliptical profile of the collimated spot, the two optical measuring device rotation angles α and β can be determined, and now the direction vector of the measuring direction of the optical measuring device is analyzed, i.e., the direction of the Z″ axis after the initial Z-axis has been transformed twice. In O-XYZ coordinates, let the initial direction vector be [0,0,1]′ by multiplying the two rotation matrices and applying to this direction vector. The direction vector for the Z″ axis can be determined.

The first transformation is a rotation of α about the y-axis and the corresponding rotation matrix is:$$R_{y}(\alpha )=\left[\begin{array}{ccc} cos(\alpha ) & 0 & -sin(\alpha )\\ 0 & 1 & 0\\ sin(\alpha ) & 0 & cos(\alpha ) \end{array}\right]$$

The second transformation is a rotation around the X′ axis by β following the initial transformation, with the corresponding rotation matrix being:$$R_{x}(\beta )=\left[\begin{array}{ccc} 1 & 0 & 0\\ 0 & cos(\beta ) & sin(\beta )\\ 0 & -sin(\beta ) & cos(\beta ) \end{array}\right]$$

By multiplying these two rotation matrices and applying them to the initial direction vector of the Z-axis, the direction vector of the Z″ axis can be ascertained:$$z\prime \prime =\left[\begin{array}{ccc} 1 & 0 & 0\\ 0 & cos(\beta ) & sin(\beta )\\ 0 & -sin(\beta ) & cos(\beta ) \end{array}\right]\left[\begin{array}{ccc} cos(\alpha ) & 0 & -sin(\alpha )\\ 0 & 1 & 0\\ sin(\alpha ) & 0 & cos(\alpha ) \end{array}\right]\left[\begin{array}{l} 0\\ 0\\ 1 \end{array}\right]=\left[\begin{array}{c} -sin(\alpha )\\ cos(\alpha )sin(\beta )\\ cos(\alpha )cos(\beta ) \end{array}\right]$$

This is the direction vector of the Z″ axis after two transformations in the initial coordinate system of the collimated beam, which is both the measurement direction vector of the optical measurement device and the measurement direction vector of the ring beam cross-section. In the process of 3D reconstruction of the cross-sectional point cloud, the parameters of the measurement direction vector of each cross-sectional measurement are introduced, and the spatial coordinates of each cross-sectional point cloud are multiplied by the corresponding rotation matrix, which can generate the calibrated spatial point cloud coordinates:(5)$$\left[\begin{array}{l} x_{i}^{\prime }\\ y_{i}^{\prime }\\ z_{i}^{\prime } \end{array}\right]=\left[\begin{array}{ccc} 1 & 0 & 0\\ 0 & cos(\beta ) & sin(\beta )\\ 0 & -sin(\beta ) & cos(\beta ) \end{array}\right]\left[\begin{array}{ccc} \cos\left(\alpha \right) & 0 & -\sin\left(\alpha \right)\\ 0 & 1 & 0\\ \sin\left(\alpha \right) & 0 & \cos\left(\alpha \right) \end{array}\right]\left[\begin{array}{l} x_{i}\\ y_{i}\\ z_{i} \end{array}\right]$$
where x_i_ and y_i_ represent the coordinate data sets of the cross-section point clouds, and z_i_ represents the cross-section sampling interval, with i = 1, 2, 3, …, *n*.

In the 3D reconstruction of the cross-sectional point clouds, the initial step involves introducing the positional changes of the collimated spot center coordinates to compensate for the point cloud stitching errors caused by the radial offset of the optical measurement device. Subsequently, the measurement direction vector Z″ of the ring beam cross-section is introduced to compensate for the point cloud stitching errors due to the angular deviation of the optical measurement device. Following these two corrective steps for the cross-sectional point cloud, the spatial posture and position at the moment of ring beam cross-sectional acquisition can be accurately reproduced.

## 4. Experiment and Analysis

### 4.1. Ring Gauge Calibration Experiment

To verify the accuracy and reliability of the point cloud calibration theory in absolute dimension measurement, this paper adopts Zhang’s calibration method [34] for system calibration. Ring gauges, as standards, are highly consistent in size and shape and can provide a reliable standard to verify the accuracy of a proposed theory. A 120.00 mm ring gauge produced by Mitutoyo Corporation is utilized as the reference standard. The schematic diagram of the ring gauge detection by the ring beam measurement device is depicted in Figure 7. Figure 7a illustrates the image of the ring beam and collimated spot captured by the industrial camera, and Figure 7b displays the schematic diagram of the ring gauge measurement experiment. In the figure, the optical measurement device is mounted at the end of an electric cylinder, the ring gauge is positioned on a V-block, the collimated beam intersects with the light screen to produce a collimated spot, and the light plane intersects with the inner wall of the ring gauge to generate a ring beam.

The cross-sectional sampling spacing is set to 1 mm, 20 cross-sectional point cloud data of the ring gauge are extracted, 191,840 point cloud coordinates are compared and analyzed with the standard model, and the tolerance range is set to ±0.01. Figure 8a shows the 3D comparative analysis of equally spaced splicing of cross-sectional point clouds, and lists the results of the distribution histograms of the distance from each point cloud coordinate point to the outer surface of the standard model, the standard deviation, the percentage of point clouds within the tolerance, and the percentage of point clouds outside the tolerance, where the percentage of key parameters outside of the tolerance range is 6.752%. The theory in Section 3.3 is applied to calibrate the cross-sectional point cloud, and the 3D comparison cloud image after calibration is shown in Figure 8b. The experimental comparison results show that the 3D reconstruction data calibrated by the cross-sectional point cloud are closer to the true value of the ring gauge, and the calibration effect is obvious, in which the percentage of the point cloud outside of the tolerance range is reduced from 6.752% to 0.955%, and the average relative error is reduced from 0.075% to 0.014%.

Through the ring gauge point cloud 3D comparison experiment, this study initially verifies the effectiveness of the cross-sectional point cloud calibration theory; however, the ring gauge does not belong to the deep hole workpiece. In order to verify the reliability of the point cloud calibration effect, it is necessary to further measure and analyze the deep hole parts, and to take the “percentage of the point cloud outside of the tolerance range” as an indicator of the evaluation of the effectiveness of the point cloud calibration.

### 4.2. Deep Hole Measurement Experiment

As illustrated in Figure 9, an experimental platform is constructed based on the detection principle, with the optical measurement device mounted at the end of an electric telescopic cylinder. A seamless steel pipe, with a nominal diameter of 120 mm and a length of 1300 mm, is positioned on a V-block. The ring beam generated by the annular laser intersects with the hole wall, producing a ring beam that, along with the collimated spot generated by the collimated laser on the light screen, is clearly visible. Initially, the collimated beam is projected onto the light screen to form a collimated spot, and the angle of the collimated beam is adjusted. By fine-tuning the four-dimensional adjustment frame of the laser, the collimated spot is adjusted to a standard circle. This adjustment indicates that the collimated beam is parallel to the light screen, which means the collimated laser beam is parallel to the optical measurement device, thus meeting the ideal measurement conditions for the initial position. To enhance the calibration effect, the measurement experiment is conducted within the range where the cantilever of the electric cylinder varies significantly, setting the travel between 1000 mm and 1500 mm for deep hole measurement.

The sampling interval is set to 1 mm, and the point cloud for the collimated spot contour of each cross-section is extracted. The theory described in Section 3.3.1 is applied to fit an ellipse to the coordinates of the collimated spot profile and to analyze the variation patterns of its ellipse parameters. Figure 10 illustrates the variation patterns of the ellipse center coordinates, with Figure 10a depicting the changes in the x-coordinate of the ellipse center and Figure 10b depicting the changes in the y-coordinate of the ellipse center. As shown in the figures, the x-coordinate value of the ellipse center remains essentially unchanged, while the y-coordinate value of the ellipse center exhibits a gradually increasing trend.

By calculating the elliptical parameters of the collimated spot, the rotation angle of the optical measurement device can be determined. Figure 11 illustrates the variation patterns of the angular deviation of the optical measurement device, where Figure 11a depicts the variation in the yaw angle α, and Figure 11b depicts the variation in the pitch angle β. As evident from the figure, the yaw angle α remains largely unchanged, while the pitch angle β exhibits a gradually increasing trend.

In summary, by fitting ellipses to the contours of 500 collimated spots and analyzing the variation patterns of their geometric parameters, the experimental data suggest that the spatial position and posture of the optical measurement device have undergone changes. Its motion trajectory approximates the deflection curve equation of a cantilever beam.

By processing data from 500 ring beam and collimated spot images, 500 sets of ring light cross-sectional contour point cloud coordinates and collimated spot contour point cloud coordinates are obtained. The variation patterns of the shape and position of the collimated spot contour have been analyzed above. Now, a comparison and analysis of the 3D reconstruction of the inner surface of the deep hole is conducted. In order to make the calibration effect more convincing, this paper adopts the deep hole inner surface point cloud coordinates and model comparison to verify the calibration effect. Due to the measured object for the steel pipe, its manufacturing precision is not high, so the tolerance range of ±0.1 mm is set for 3D point cloud comparison experiments. Figure 12 shows the 3D comparison cloud images of the cross-sectional point cloud, both calibrated and uncalibrated, within the tolerance range of ±0.1 mm. Figure 12a presents the 3D comparison cloud plot of the cross-sectional point cloud with isospace point cloud stitching, where 8.091% of the point cloud is outside of the tolerance range. Based on the equally spaced point cloud stitching, the first point cloud calibration is performed by introducing the radial offset of the optical measuring device to the coordinates of each cross-sectional point cloud. Subsequently, the measurement direction vector of each cross-section is introduced based on the first calibration, and the point cloud of each cross-section is transformed by spatial coordinates so that the normal vector of the point cloud plane coincides with the measurement direction vector, completing the secondary point cloud alignment. Figure 12b presents the 3D comparison cloud image of the cross-sectional point cloud after calibration, where 1.804% of the point cloud is outside of the tolerance range.

Compared with the ring gauge experiment, the deep hole measurement experiment is conducted on an actual workpiece, where the measurement environment is more complex. This allows for a more comprehensive evaluation of the technology’s practical application. Statistically, 4,867,124 internal surface point cloud coordinates of the deep hole workpiece are collected in this experiment. The 3D comparison of the internal surface point clouds of the deep hole is accomplished using equally spaced cross-sectional point cloud stitching and point cloud calibration stitching, respectively. The experimental results demonstrate that the cross-sectional point cloud calibration is significant, reducing the percentage of point cloud data outside of the tolerance range from 8.091% to 1.804%. Consequently, the accuracy of the 3D reconstruction model is improved by 6.287%. The validity and reliability of the annular optical cross-sectional point cloud calibration theory proposed in this paper are further verified through deep hole measurement experiments.

## 5. Discussion and Conclusions

This paper proposes a ring light point cloud calibration technique based on collimated laser beams for calibrating point clouds of deep hole inner surfaces. We verified the accuracy and practicality of this technique through ring gauge experiments and deep hole measurement experiments. The ring gauge experiment provided a standardized validation method for this study. The calibrated point cloud model was significantly closer to the ideal state compared with the uncalibrated model, providing an initial assessment of the technology’s performance. The deep hole measurement experiment demonstrated the advantages of this technology in practical applications. The experimental results show that the proposed ring light cross-section calibration technique based on collimated laser beams can significantly reduce the proportion of point cloud data exceeding the tolerance range, improving the accuracy of 3D reconstruction of deep hole inner surfaces by 6.287%.

Compared with existing calibration methods [14,16,35], the point cloud correction method proposed in this study has significant advantages. Based on traditional ring light measurement devices, this study innovatively introduces collimated light beams as a reference baseline. The industrial camera can simultaneously capture images of the ring light beam and the collimated light spot. By analyzing the elliptical shape and position changes of the collimated light spot, the spatial pose of the ring light measurement device can be calculated in real time, thus providing accurate correction parameters for the ring light cross-sectional point cloud. Furthermore, the calibration process of this method has real-time dynamic characteristics, which can effectively address environmental vibration issues common in industrial applications. Even when the optical measurement device is swinging, this technology can quickly and accurately restore the spatial position of the ring light cross-section at the moment of camera capture, thereby compensating for errors caused by vibration and ensuring high precision of measurement data.

This technology not only enhances the accuracy of data acquisition but also provides an efficient and reliable means for inspecting the interiors of deep holes. It is particularly suitable for aerospace, automotive, national defense, and other fields requiring extremely high precision. In the future, this study plans to integrate the optical measuring device with a pipeline robot, replacing the electric cylinder with a pipeline robot drive, and using the collimated laser beam as the measurement reference. This will enable the inspection of inner walls of deep holes with very large depth-to-diameter ratios. Additionally, ongoing research will focus on optimizing the technology, expanding its applications, integrating equipment design, and enhancing the intelligence of the measuring system. These efforts aim to further improve the system’s responsiveness and ease of operation, promoting its application in a broader range of industrial scenarios.

## Figures and Tables

**Figure 1 sensors-24-05790-f001:**
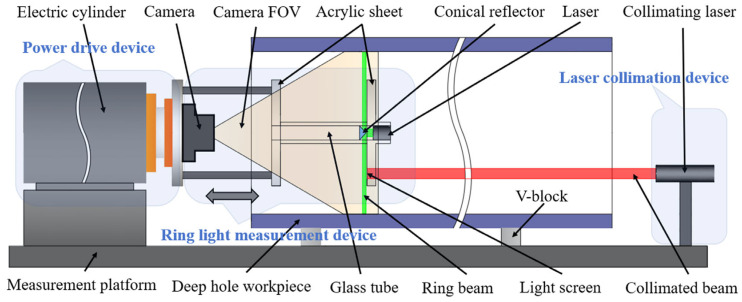
Diagram of deep hole detection principles.

**Figure 2 sensors-24-05790-f002:**
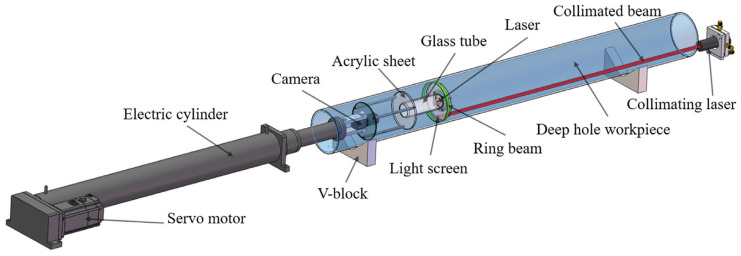
Structure diagram of deep hole detection system.

**Figure 3 sensors-24-05790-f003:**
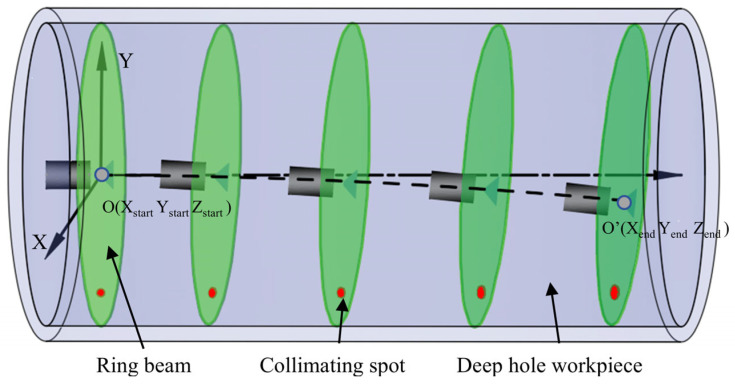
Ring beam scanning trajectory.

**Figure 4 sensors-24-05790-f004:**
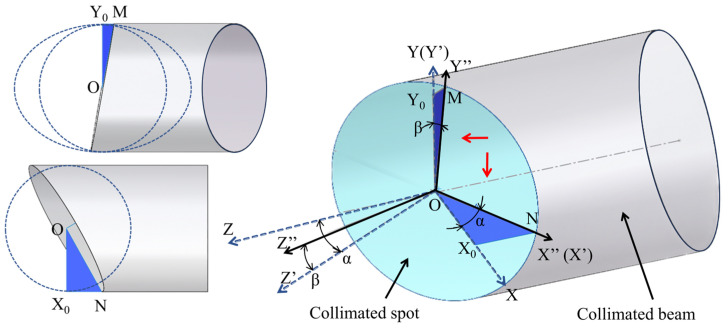
Schematic diagram of the variation of the collimated spot profile caused by the spatial attitude change of the optical measuring device.

**Figure 5 sensors-24-05790-f005:**
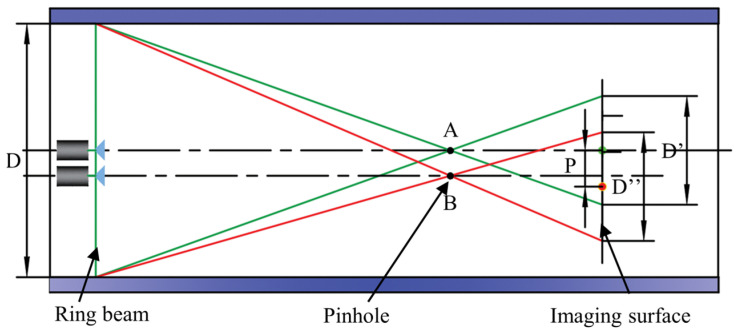
Schematic diagram of the camera pinhole imaging model. (Green line: ring beam pinhole imaging model when the optical measuring device is in the initial position. Red line: ring beam pinhole imaging model when the optical measuring device deviates from the initial position.)

**Figure 6 sensors-24-05790-f006:**
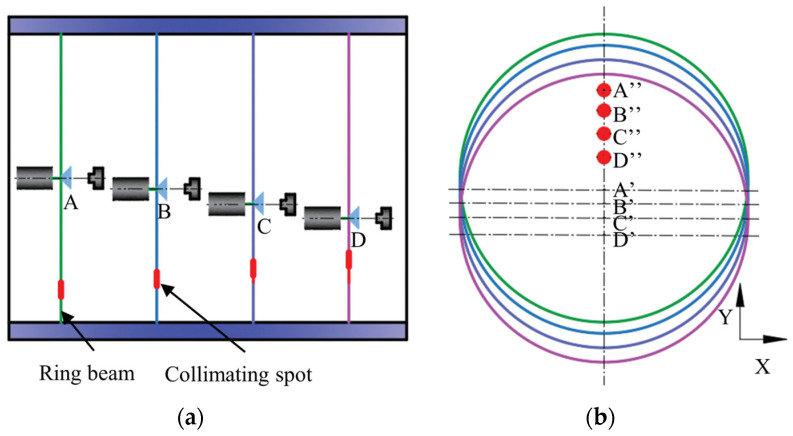
Schematic diagram of the change in the coordinates of the ring beam contour due to the radial offset of the optical measurement device. (**a**) Schematic diagram of the radial offset of the optical measuring device; (**b**) trajectory of the change of the ring beam and collimated spot in the camera viewpoint.

**Figure 7 sensors-24-05790-f007:**
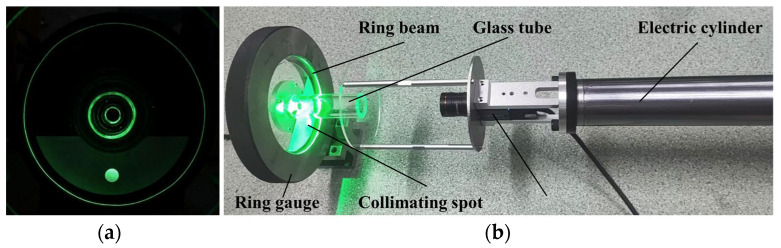
Ring gauge detection experiment. (**a**) Image of the ring beam and collimated spot; (**b**) ring gauge experimental platform.

**Figure 8 sensors-24-05790-f008:**
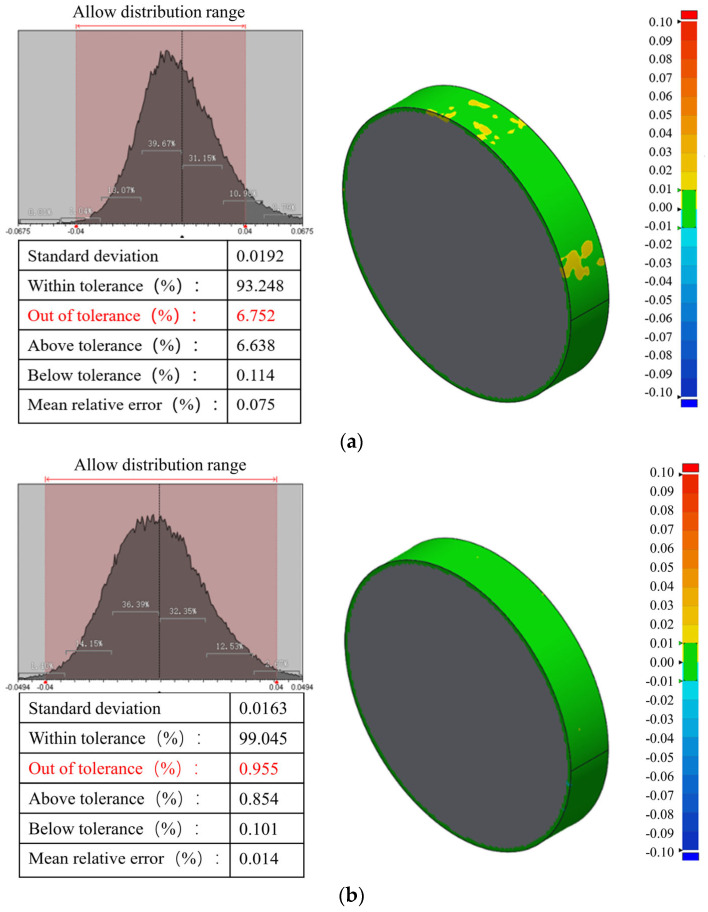
Three-dimensional comparison of calibrated cross-sectional point clouds of the ring gauge: (**a**) three-dimensional comparison of equidistantly stitched cross-sectional point clouds; (**b**) three-dimensional comparison of calibrated cross-sectional point clouds.

**Figure 9 sensors-24-05790-f009:**
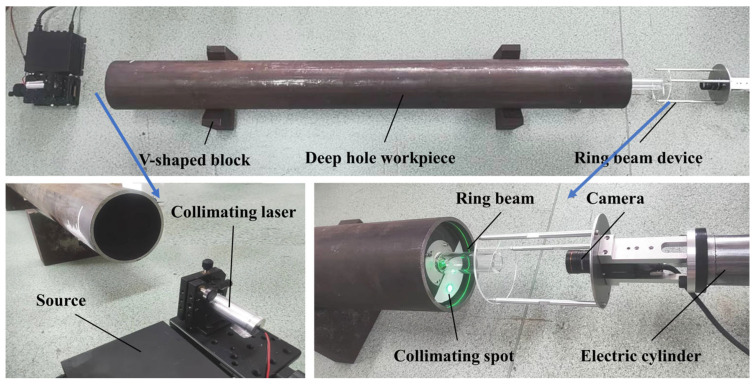
Deep hole detection experimental platform.

**Figure 10 sensors-24-05790-f010:**
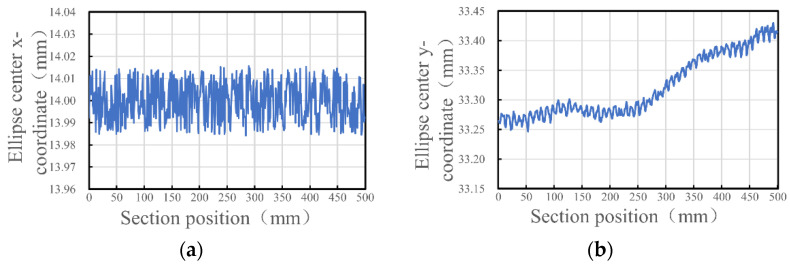
Trends in the center coordinates of the collimated light spot. (**a**) Trend in the x-coordinate of the ellipse center; (**b**) trend in the y-coordinate of the ellipse center.

**Figure 11 sensors-24-05790-f011:**
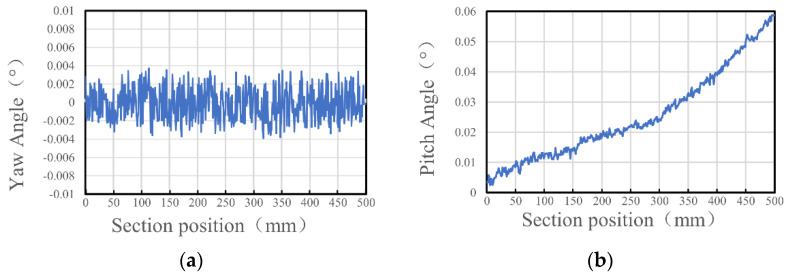
Trends in angular deviation of the optical measuring device. (**a**) Trend in the yaw angle α; (**b**) trend in the pitch angle β.

**Figure 12 sensors-24-05790-f012:**
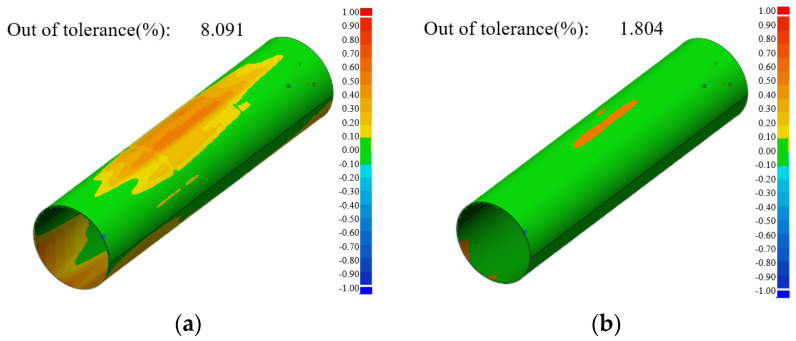
The 3D comparison cloud map of the inner surface of the deep hole (tolerance: ±0.1 mm): (**a**) 3D comparison cloud of point clouds of equally spaced spliced ring optical cross-sections; (**b**) 3D comparison cloud of calibrated point clouds of ring optical cross-sections.

## Data Availability

Data are contained within the article.

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
