# Peer review of "Research on Point Cloud Acquisition and Calibration of Deep Hole Inner Surfaces Based on Collimated Ring Laser Beams"

_sensors, 2024, doi:10.3390/s24175790_

Round 1

Reviewer 1 Report

Comments and Suggestions for Authors

1) The refs 7 and 8 the same. They have identical title.

2) Moreover, all refs have double numbers. Please correct it.

3) The literature review is very narrow and includes only 22 refrences. To make the paper more interesting to broad auditory I recommend expanding it with the description of more broad range of related solutions, like the following:

- Noncontact holographic technique for the measurement of cylindrical shapes: https://doi.org/10.1117/1.OE.55.12.121719

- Digital holographic system for straight pipes inspection,

  where the defects measurements in high height range is implemented by means of synthetic wavelength from multicolor light source  https://doi.org/10.1016/j.optlaseng.2017.05.012

- Digital holography, where the defects measurements in high height range is implemented by means of variation of the tilt angle illumination https://doi.org/10.1117/1.OE.59.10.102414

- Confocal microsopy for hole internal shape characterization during the laser percussion drilling https://doi.org/10.1016/j.ijmachtools.2013.08.004

- etc.

4) "ring light measurement device, power drive device and laser collimation device." Please, show these three parts on Fig. 1.

5) "The ring light source is irradiated by a semiconductor laser beam" - it is difficult to locate it, since the black line, aimed to designate it is too short.  And it is unclear how the green light affter conical mirror reflected to the camera direction from the blue edges (BTW, I have no idea, what these edges mean)?

6) "The drive device is a multi-section motorized telescopic cylinder" is unclear. Where is it? Where the motor?

7) Describe the role of light screen in details. Is it transparent? It looks like beam steering device, which directed transmitted light to the camera. But I suppose, it is not so.   

7) Describe, how the light source initially centered?

8) "At the initial position, the optical axis of the optical measuring 128 device coincides with the deep hole axis..." Where the deep hole there? I'm sorry, but I do not understand. It is not shown neither on Fig. 1 nor Fig. 2.

9) "Utilizing the point cloud data, scientific algorithms enable the derivation of geometric parameters such as the inner diameter, roundness, and straightness of the deep hole, in addition to information on internal surface defects." I'm not sure, that you able to measure all these parameters. You not explained the method clearly. What happens, if the position of the irradiating laser will be shifted during the measurements? 

10) The text should be carefully checked. Coorect the misprints, wrong placement of spaces (like in Figure 5 caption: "...Device ;(B)..."), etc.

11) Sometimes, it is seems, that the work was written by GPT-chat. It lacks of clarity, and contains such strange words, like "scientific algorithms'. Please explain the method more clearly. This is main comment, which requires major revision.

Comments on the Quality of English Language

The style of narration should be improved. Sometimes. it is seems, that the work was written by GPT-chat. It lacks of clarity, and contains such strange words, like "scientific algorithms'. etc...

Author Response

Hello reviewer, please see the attachment.

Reviewer 2 Report

Comments and Suggestions for Authors

In this paper, the authors demonstrated experimentally that using an additional collimated reference light beam in addition to the annular light beam allows the number of points outside the tolerance to be reduced from 8% to 2%. The optical scheme for determining the inner surface of pipes with a diameter of about 100 mm with an accuracy of about 100 μm (Fig. 1) is novel due to the laser with a collimated light source. Therefore, the work can be published after the authors take into account the comments.

Comments

1. In equation (3), bc should be replaced by cy^2.

2. Lines 430 and 431: two identical sentences.

3. Line 430 says that the tolerance range is 0.1 mm, while in Fig. 9 the measurement accuracy is 0.01 mm. It is necessary to explain why the tolerance parameter is used if the deviation from cylindricity measured is 10 times smaller than the tolerance allows.

4. The authors showed that they reduced the number of measurement points that go beyond the tolerance value from 8% to 2%. However, the paper does not indicate the accuracy of this method, that is, what is the minimum deviation from the cylindrical shape of the pipe that can be detected. This data should be provided.

5. The paper also does not compare the accuracy with other similar methods. For example, in Opt Express 27, 29697 (2019), the error in determining the 3D surface relief is 0.01%. The authors should compare the accuracy of the developed method with other known methods in the Conclusion. Until this is done, it cannot be said that this method is more efficient and accurate than other known methods.

6. The bibliography has double numbering.

Author Response

(The authors gave the same response as above.)

Round 2

Reviewer 1 Report

Comments and Suggestions for Authors

Dear Authors,

Thank you for the improvement of the manuscript. The quality of the work was improved seriously. But there following issues still should be addressed:

1. The reference list was improved seriously. While the work was supplemented by relevant references, I found, that some of the works has not been cited. I suppose, that you came to the conclusion, that  these works not directly related to the discussion. Therefore, I would like to give an additional clarification, how they related to the problem being discussed and how the can improve the manuscript.

1.1. You included ref. [26], where multiwavelength color phase-shifting digital holography was implemented. The measurements of the surface profile of the pipe inner wall in the heights range, exceeding the single laser wavelengths is implemented there through the synthetic wavelength. The next from the proposed references (https://doi.org/10.1117/1.OE.59.10.102414), is not directly related to the measurement of the pipes. However, this work describes the tilt angle illumination approach, which is also aimed to improve the heights measurement range, and represent an effective alternative to the synthetic wavelengths measurements, which requires the very stable two-wavelengths source. So, the following clarification can be made: "Yokota [26] also used holographic interferometry to measure the inner wall of a pipeline and applied synthetic wavelength to perform measurements in the heights range, exceeding the single laser wavelengths. This approach requires the very stable two-wavelengths source, so the tilt angle illumination method can be an effective alternative [https://doi.org/10.1117/1.OE.59.10.102414]. These methods can achieve high accuracy ..." I believe, that holographic techniques very effective in the solution of the problem of pipe inspection. But indeed, sometimes they are complicated enough. That is why, mentioning the much more simple alternatives can be useful for the development of this research topic. 

1.2. The work, describing the application of the confocal microsopy for hole internal shape characterization during the laser percussion drilling [https://doi.org/10.1016/j.ijmachtools.2013.08.004] was recommended for the consideration, because it can brighten up the general discussion in the beginning of the introduction about the relevance of deep hole detection. Thus, precise detection of internal structures in deep holes not only imprtant for the inspection of the engine and hydraulic cylinders, pipelines, and artillery barrels, but also take place in the optimizing of the laser percussion drilling, which in turn have the applications, e.g. for the creating micro-holes in nickel-based alloy blades for cooling of aircraft turbines. I believe, this extention of the description of the relevance of the problem will make the paper more interesting for the broad audithory.

2. In the response on the comment 5 you added the text to the manuscript, which includes the following fragment (lines 123-125): "The blue part in Figure 1 represents the deep hole workpiece. When the 360-degree omnidirectional laser beam intersects with the inner wall of the deep hole, it generates a ring beam." First of all, which blue part in Fig. 1 do you mean? there are several of them. Is it "Ring light measurement device"? Secondly, it is not explained, that the "ring beam" is. It seems, that it is not conventional laser beam. Instead, it is a radiation scattered on the inner wall of the pipe. And as a scattered radiation, it is propagates to the multiple of directions. And some part of this radiation is propagated to the camera direction. Am I right? When you wrote "ring laser beam", this term is confusing, since it close to the vortex beams. So, please correct the methodology and add proper explanation to the text of the manuscript. "Camera view" can be replaced with "Camera FOV" (if you will introduce such an abbreviation in the text).    

3. I recommend including to the text of the manuscript the clarifications, given to the Comment 6. Even the image can be useful for the readers, to imagine the whole device.

4. Laser radiation should produce speckle patterns. Can you add discussion of this isue to the manuscript. Why it is not impact the measurement prosess?

5. In continue of the discussion, started in Comment 8: I think it would be more correct to title your pipe as not a deep hole. In fact, you inspect "inner walls of the pipe".

6. The answer on the question in Comment 9 ("If the position of the projected laser beam shifts during the measurement process, it indicates that the spatial position and orientation of the optical measurement device have changed") can be included to the text of the manuscript.

And finally, thank you for the honest description of using GPT chat. I think, this can be reflected at the end of the article. And please, pay more attention on the terminology. I mean the comments 2 and 5 of this review, in particular, but there also small flaws in general.

Author Response

(The authors gave the same response as above.)

Reviewer 2 Report

Comments and Suggestions for Authors

The authors took into account all my comments and therefore the work can be published.

Author Response

Dear Reviewer,

Thank you for your review and feedback on my paper. I am pleased to hear that you agree to publish my paper, and I greatly appreciate that you considered all the points I raised during the revision process. Your comments have played a crucial role in improving my work.

Once again, thank you for your support and encouragement.

Best regards,

[Huifu Du]